# Accelerometer-Measured Physical Activity Levels and Patterns Vary in an Age- and Sex-Dependent Fashion among Finnish Children and Adolescents

**DOI:** 10.3390/ijerph19116950

**Published:** 2022-06-06

**Authors:** Anne-Mari Jussila, Pauliina Husu, Henri Vähä-Ypyä, Kari Tokola, Sami Kokko, Harri Sievänen, Tommi Vasankari

**Affiliations:** 1The UKK Institute for Health Promotion Research, Kaupinpuistonkatu 1, FI-33500 Tampere, Finland; anne-mari.jussila@elisanet.fi (A.-M.J.); pauliina.husu@ukkinstituutti.fi (P.H.); henri.vaha-ypya@ukkinstituutti.fi (H.V.-Y.); kari.tokola@ukkinstituutti.fi (K.T.); harri.sievanen@ukkinstituutti.fi (H.S.); 2Faculty of Sport and Health Sciences, The University of Jyväskylä, Seminaarinkatu 15, FI-40014 Jyväskylä, Finland; sami.p.kokko@jyu.fi; 3Faculty of Medicine and Health Technology, Tampere University, Kalevantie 4, FI-33014 Tampere, Finland

**Keywords:** physical activity, sedentary behavior, daily steps, moderate intensity, light intensity

## Abstract

Background: The purpose of this study was to measure physical activity (PA), sedentary behavior (SB), and hour-by-hour PA patterns with an accelerometer in a population-based sample of Finnish children and adolescents. Methods: A total of 3274 participants (3rd, 5th, 7th, 9th graders) from 176 schools wore a hip-worn triaxial accelerometer for seven days during waking hours. Mean amplitude deviation of the acceleration data was used to assess PA intensity that was converted to metabolic equivalents and categorized into light, moderate, and vigorous PA. Angle for posture estimation was used to measure SB and standing. Results: The majority of participants’ PA consisted of light PA, and they were sedentary for more than half of their waking hours. Children were more active than adolescents, and boys were more active than girls. Participants took, on average, 9890 steps daily, and one third met the PA recommendation. The participants were divided into tertiles based on daily steps to investigate the variation in PA patterns. Compared to the least active tertile, the most active tertile took twice as many steps on weekdays and nearly three times as many steps on the weekend. Conclusions: The majority of the participants were not active enough, and there was a great variation in PA levels and patterns, especially among the adolescents and on weekends.

## 1. Introduction

Physical activity (PA) is important for children and adolescents to enhance growth and health [1,2]. At the same time, a high amount of sedentary behavior (SB) is harmful for health [1,3]. According to accelerometer-based measurements, most children’s and adolescents’ PA consists of light physical activity (LPA) [4,5]. Approximately half of children and less than one-quarter of adolescents meet the PA recommendation [1,6], i.e., at least 60 min moderate-to-vigorous physical activity (MVPA) in a day [4,5,6,7,8]. In addition, children and adolescents are sedentary for most of their waking hours [4,5,7,8]. 

PA is a multi-dimensional construct, and no single measure can assess all its facets [9]. In contrast to subjective self-reports, accelerometer measurements currently represent the most accurate means for assessing PA and SB [10,11]. With these measurements, we can get specific information about the intensity of PA, e.g., peak cadence [12], the different profiles on daily PA and SB, [13,14], and the accurate MVPA amounts expressed in metabolic equivalents (MET) [11]. 

Daily step count is one indicator of PA. In adults, daily step counts less than 5000 steps per day have been used to define a sedentary lifestyle, whereas step counts exceeding 10,000 steps per day indicate a physically active lifestyle [15]. There is still little evidence to advocate specific cut-points for children and adolescents, although a few alternative values have been considered [14,15]. Tudor-Locke et al. [16] have suggested that to achieve the recommended amount of PA in school-aged children, boys should take 13,000–15,000 steps daily, and girls should take 11,000–12,000 steps daily. However, the evidence behind these amounts is speculative [16].

Both subjective and device-based measurements have shown that younger children are, on average, more active than adolescents [4,5,7,8,13]. In general, boys seem to be more active than girls, and weekdays include more PA than weekends in both sexes [4,5,7,8,13]. In Finland, the trends in children’s and adolescents’ PA and SB are similar to those observed in other Western countries [5,17,18,19,20].

The purpose of the present population-based study was to describe accelerometer-measured PA and SB among a large sample of Finnish children and adolescents. In addition, we analyzed PA according to step levels and how many of the participants met the PA recommendations. A further aim was to examine hour-by-hour patterns of PA and SB during waking hours on weekdays and weekend days, and to make age and sex group comparisons of the daily PA patterns.

## 2. Materials and Methods

### 2.1. Study Design and Sample Size

This study is based on the Finnish School-age Physical Activity (FSPA) 2016 study on children’s and adolescents’ PA and SB. This national study was carried out for the first time in 2014 as a survey in connection with the World Health Organization (WHO) Cross-national Health Behavior in School-aged Children (HBSC) survey [18]. In 2016, the FSPA study was carried out independently during the spring term from February to May, and the study sample included 3rd, 5th, 7th, and 9th graders attending normal education in public schools. In the Finnish school system, these pupils are generally 9-, 11-, 13-, and 15-years-old, respectively. In this study, ages of 9–11 years old are considered children, and 13–15 years old are considered adolescents. In 2016, the accelerometer-based measurements were carried out in parallel with the survey for the first time. The research responsibility of the survey in the FSPA study was at the University of Jyväskylä, whereas the UKK Institute was responsible for accelerometer-based measurements. Ten regional organizations (Figure 1) executed the measurements under the supervision of the UKK Institute.

A random sample of schools was drawn from the Statistics Finland database based on the HBSC protocol [21] for the FSPA 2016 survey data collection. The accelerometer measurements of PA and SB were conducted in a sub-sample of the survey sample. The inclusion criterion for schools involved was a maximum 100-km distance between the school and regional co-operative organizations (Figure 1).

A total of 285 schools with 10,513 possible participants were willing to participate in the FSPA study’s online survey, whereas 176 of those schools with 7061 possible participants were willing to participate in the accelerometer measurements. Altogether, 3274 children and adolescents (57% girls) used the accelerometer (Table 1). The inclusion criterion to the present study was that the participants had used the accelerometer for at least four days (including one weekend day) with a minimum of 10-h daily wearing time for one week (seven consecutive days). Eligible accelerometer data were obtained from 2981 children and adolescents (91% of participants). Younger children took part in the study more actively than the older ones, and girls more actively than boys (Table 1).

### 2.2. Accelerometer-Based Measurements of PA and SB

The accelerometer-measured PA and SB data were collected with triaxial hip-worn UKK AM30 and RM42 accelerometers (UKK Terveyspalvelut Oy, Tampere, Finland), which have been validated in adolescents [22] and adults [23,24,25]. The latter accelerometer mentioned is an updated version of the first one. Both accelerometers employ the same sensor components, collect the data with the same specifications, and the data analysis algorithms are identical. 

Study coordinators in the ten regional organizations, together with local school teachers, delivered the accelerometers during school hours, and gave both oral and written instructions to the children and adolescents about the proper use of the device. One week later, the devices were returned to the schools. The participants were instructed to wear the accelerometer on an elastic belt on the right side of their hip for seven consecutive days during waking hours. The participants were advised to remove the device during shower or water-based activities. 

Raw triaxial acceleration data were collected at 100 Hz sampling frequency. The data were analyzed with validated algorithms based on the mean amplitude deviation (MAD) of the acceleration signal analyzed in 6-s epochs and angle for posture estimation (APE) [23,24,25]. The MAD value corresponds to the intensity of PA, whereas APE recognizes the body posture (i.e., lying, sitting, and standing) following the recent recommendations [26]. The MAD values were converted to METs [24]. The epoch-wise MET values were further smoothed by calculating a one-minute exponential moving average. Using the smoothed MET values, the total PA was classified as light (1.5–2.9 MET), moderate (3.0–5.9 MET), or vigorous (6 MET or more) activity. In the final results, MVPA was combined, since vigorous PA (VPA) covered a very small proportion of the total wearing time. Lying and sitting denoted SB, whereas standing was analyzed separately. 

Appendix A shows the relationship between oxygen consumption and MAD values for adults and children to demonstrate the validity of the methods. The adult results are from 29 (14 female + 15 male) participants aged from 20 to 59 years [24]. The children’s results are from 34 (21 girls + 13 boys) participants aged from 7 to 11 years [27]. Adults performed a pace-conducted test on a track at speeds of 2.2 km/h, 3.6 km/h, 5.0 km/h, 6.5 km/h, 7.9 km/h, 9.4 km/h, 10.8 km/h, and 12.2 km/h. The children’s values are from the lowest to the highest intensity: walking 4 km/h on the treadmill, walking at a self-selected speed on a track, walking 6 km/h on the treadmill, running 8 km/h on the treadmill, and running at a self-selected speed on a track.

The steps were identified from the vertical component of the acceleration with the algorithm proposed by Vähä-Ypyä et al. [23].

### 2.3. Statistical Analyses

Mean and standard deviation (SD) are shown separately for boys and girls by the school grade. The Mann–Whitney U test was used to test the differences between sexes and grades. In order to study the variation in PA and SB patterns within the age groups, children and adolescents were divided into sex- and grade-specific tertiles (most active, middle, and least active group) based on their daily step counts. In this study, these tertiles are called simply ‘tertiles’. The Kruskal–Wallis H test was used to test the differences between the tertiles for hour-by-hour data. Sex and grade differences for meeting the PA recommendation were analyzed by logistic regression. Inverse probability weighting was used to correct the differences in area, sex, and age between the sample and population. The significance level was set at *p* < 0.05. The Benjamini–Hochberg procedure [28] for adjusting the significance level of *p*-values for all 354 statistical tests in this manuscript resulted in an adjusted significance level of *p* = 0.044. All *p*-values < 0.044 were considered statistically significant. All analyses were conducted by IBM SPSS version 24.0 (IBM Corp. Armonk, NY, USA).

### 2.4. Ethics

Children, adolescents, and their guardians were informed about the FSPA study beforehand via schools. Despite the school-level consent for participation, individual participation was completely voluntary. Written informed consent was obtained from children’s and adolescents’ guardians before the accelerometer measurements. Participants had a right to withdraw from the study at any time without a specific reason. The study followed the ethical principles of the Declaration of Helsinki, and was approved by the Ethics Committee of the University of Jyväskylä.

## 3. Results

### 3.1. Overall PA and SB

Children’s and adolescents’ PA and SB times, and mean daily step counts during the accelerometer wearing time are presented in Table 2 by grade and sex. The mean proportions of PA and SB are shown in Figure 2.

Most of the children’s and adolescents’ daily PA consisted of LPA (on average, 3 h 31 min per day). On average, younger participants had significantly more LPA than the older ones (*p* < 0.001). Girls had significantly more LPA than boys between the 3rd and 5th grades (*p* < 0.001), but significant between-sex differences disappeared among the 7th and 9th graders.

Children and adolescents spent 8–18% of their waking hours in MVPA (on average, 1 h 46 min per day), and the levels of MVPA among the older age groups were significantly lower than those among the younger groups (*p* < 0.001). On average, boys had more MVPA than girls in every age group (*p* < 0.001). Vigorous PA (on average, 14 min per day) covered only a few percentages of the waking hours among both sexes. 

Children and adolescents were sedentary (on average, 7 h 27 min) or standing (on average, 1 h 18 min) more than half of their daily waking hours. SB increased significantly with age (*p* < 0.001), and boys were, on average, more sedentary than the girls (*p* < 0.05) in all age groups except the 7th grade. Moreover, standing increased with age. However, there was no statistically significant difference in standing between the 3rd and 5th graders (*p* = 0.562). Among adolescents, girls stood, on average, more than boys (*p* < 0.001). 

Children and adolescents took, on average, 9890 steps per day. The mean number of steps decreased almost 30% from 3rd to 9th grade (from 11,421 to 8220). All other grades differed statistically significantly from the 3rd graders (*p* < 0.001). On average, boys took significantly more steps per day than girls (*p* < 0.027).

According to accelerometer measurements, on average, one third (31.3%) of the FSPA study participants met the PA recommendation, accumulating at least 60 min of MVPA each day. A total of 45.4% of children (3rd and 5th graders) and 18.6% of adolescents (7th and 9th graders) met the PA recommendation. Girls were less likely to meet the recommendation than boys (Figure 3).

### 3.2. Tertile-Wise Daily Step Counts

Tertile-wise daily steps broken down by sex and grade are shown in Table 3.

All the tertiles differed statistically significantly (*p* < 0.001) from each other when analyzing the differences between grades and sex. Figure 4 shows that girls and boys of the most active tertile achieved, on average, over 10,000 steps at every grade both on weekdays and weekends. On weekdays, the middle tertile of the boys in 3rd and 5th grades and the middle tertile of the girls in 3rd grade achieved this limit as well. On weekends, only the boys of the middle tertiles in 3rd and 5th grade achieved, on average, over 10,000 steps. The least active tertiles of both sexes did not reach the 10,000 steps on weekdays nor weekends. On weekends, the mean daily step count among the least active tertiles of boys and girls in the 9th grade remained less than 5000 steps. 

### 3.3. Hour-by-Hour Step Patterns

When looking at the tertiles hour-by-hour (Figure 5a) and making the comparison within grades and between sexes, the most active tertiles among both sexes accumulated, on average, more steps during leisure time (after school) on weekdays, whereas the least active tertiles accumulated most of their daily steps during school hours.

Among the most active boys, on average, the peak step count occurred at 6–7 p.m. on weekdays, with a mean of 1647 steps per hour (Appendix A). Moreover, the middle tertile had the most active hour at the same time, with a mean of 1136 steps per hour. Among girls, the two most active tertiles had the highest PA time slot at the same time as boys. However, girls accumulated lower mean step counts. The most active girl tertile achieved 1341 steps, and the middle tertile achieved 813 steps per hour. The least active tertiles of both sexes, in turn, achieved the peak step count on weekdays between 1–3 p.m. (boys with 723, and girls with 616 steps per hour). Regarding the tertiles on weekdays, there were statistically significant differences (*p* < 0.05) between mean hourly step counts in all hours between 8 a.m. to 11 p.m. (Appendix A).

Figure 5a also shows that, on weekends, all the hour-by-hour profiles among children and adolescents were flatter, and the steps accumulated more constantly during the day among every tertile. When comparing the same tertiles between sexes, boys accumulated, on average, more steps than girls during weekends in 3rd and 5th grades. Moreover, in 7th grade, the least active tertiles, and in 9th grade, the middle and the least active tertiles, did not differ from each other by sex. Similarly to weekdays, on weekends, the number of steps declined with grade among both sexes. In the least active tertiles, the highest mean step count per hour was 527 among boys, and 507 among girls (Appendix A). The highest variation in the mean hour-wise steps between tertiles, on average, was seen on weekday evenings. Among both sexes, the mean step count per hour differed statistically significantly (*p* < 0.001) between all tertiles from 8 a.m. and 10 p.m. during both weekdays and weekends (Appendix A).

### 3.4. Hour-by-Hour PA Intensity

The tertiles of all grades were closer to each other on LPA than the step counts during weekdays, especially during school hours (Figure 5b). The middle tertile of boys had, on average, slightly more LPA than the most active tertile of boys on weekdays during multiple hours (Appendix A), whereas girls accumulated more LPA minutes per hour in all grades on their leisure time than boys. There were statistically significant differences between the tertiles in LPA minutes per hour during the weekend both in boys and girls (*p* < 0.001). On weekdays, there were statistically significant differences (*p* < 0.05) between the tertiles in LPA minutes per hour from 10 a.m. to 24 p.m. among girls, but only after school hours (from 3 p.m. onwards) among boys. 

Figure 5c shows the mean 1-min MET peak value levels per hour in the tertiles. The MET peak value level reflects children’s and adolescents’ PA intensity. During weekdays, all tertiles in the 3rd and 5th grades achieved, on average, over six MET peaks, and the 7th and 9th graders achieved over four MET peaks. Among the two youngest groups, there were two MET peaks on weekdays, one during school time, and the other during leisure time. In the 7th grade, the highest MET peak occurred in leisure time in all other groups except the least active tertile. According to Appendix A, the highest mean MET peaks occurred among the most active tertiles of boys and girls during weekdays at 1–2 p.m. (7.1 METs in boys and 6.0 METs in girls). The time slot coincides with the peaks in step counts. There was also another MET peak during weekdays in the most active tertiles at 6–8 p.m. The middle and the least active tertiles among both sexes achieved, on average, the highest MET peaks on weekday afternoons. The middle tertile of boys had a 6.7 MET peak, and the least active boys had a 6.1 MET peak. In girls, the corresponding MET peaks were 5.7 METs and 5.3 METs (Appendix A). 

On the weekends, the patterns of MET peaks were again flatter compared to weekdays (Figure 5c). All tertiles in the 3rd and 5th grades achieved, on average, more than four MET peaks on weekends, whereas in older grades, the MET peaks declined to under four METs in the least active tertiles of both sexes and in the middle tertile of girls. There were statistically significant differences between the tertiles in MET peaks among both sexes between 8 a.m. and 12 p.m. (*p* < 0.05). 

## 4. Discussion

This is the first population-based study of accelerometer-measured PA and SB levels and patterns among children and adolescents in Finland presented by sex and grade. To our knowledge, this study is also the first one to compare children’s and adolescents’ hourly steps, LPA, and MET patterns in step tertiles during waking hours. With this analysis of the FSPA study, we achieved detailed knowledge about children’s and adolescents’ hour-by-hour PA and SB during weekdays and weekends.

The main results of our study indicate that PA levels vary greatly between grades and sexes. Younger children were significantly more active than adolescents, and boys were more active than the girls. Most of the children’s and adolescents’ PA consisted of LPA, and only a tiny proportion of PA was vigorous. Furthermore, the participants spent more than half of their waking hours sedentary or standing. The amount of SB increased significantly with grades, and boys were, on average, more sedentary than girls. In general, one third of all participants met the PA recommendation. The variation in meeting the recommendation between grades was significant; children met the PA recommendation more often than adolescents. Boys also met the PA recommendation more often than girls. Similar results have been reported in previous studies in other countries [4,5,6,7,29,30,31]. 

Participants’ PA patterns according to grade- and sex-specific tertiles, especially during their leisure time, varied significantly. The mean daily step count was about double on school days, and nearly triple on weekends in the most active tertile compared to the least active one. All the children and adolescents were able to gain at least a moderate PA (≥3 MET) level. On weekdays, all the tertiles of boys attained, on average, the VPA (≥6 MET) level, but among girls, only the most active tertile achieved the VPA level. On weekends, only the most active tertile of boys attained, on average, the VPA level.

Due to the great variability in measurement devices, analysis algorithms, and outcomes used in the previous PA and SB research articles [9,10,11,26,27], the comparison of the present results with previous findings must be done with caution. Husu et al. [5] used the same accelerometer and analyzing algorithms, making their results comparable with the present findings from the FSPA study. Husu et al. [5] found that in 7- to 14-year-old children, LPA covered 18% (2 h 24 min) and MVPA covered 19% (2 h 30 min) of the waking hours, and children spent, on average, 54% (7 h 18 min) of waking hours sedentary, and stood 9% of the time (1 h 15 min), which are congruent with the present study findings. Other pertinent studies have used different devices and algorithms, making the comparison between studies less straightforward. Verloigne et al. [29] reported the amount of 10–12-year-old children’s LPA being 4–5 h in five European countries. Judice et al. [30] reported in Portuguese 10−17-year-old young people, 60% (8 h 18 min) mean sedentary time, 36% (5 h 3 min) LPA time, and 5% (37 min) MVPA time of their waking hours. Roberts et al. [31] reported that the mean amount of MVPA in Canadian children aged 5−17 years was 54 min. Loyen et al. [32] stated aptly in their systematic literature review that the amount of device-based MVPA levels in European children varies a lot, from 22 min in Switzerland to 185 minutes in Estonia, and in adolescents, from 29 min in the United Kingdom (UK) to 95 min in Portugal. Steene-Johanssen et al. [33] also indicated in their harmonized analyses that children’s and adolescents’ overall PA, time spent in MVPA, sedentary time, and prevalence of being sufficiently physical active differ substantially between countries and regions.

According to the present FSPA study, younger children were more active than adolescents, and boys were, on average, more active than girls. These findings are consistent with previous studies [4,5,7,29,30,31,32,33]. The average step count in the present study was slightly under 10,000 per day, being, on average, on the same level as among the Canadian youth [34]. The step count in the FSPA study differed statistically significantly between grades and sexes. The average step count among boys declined by around 4000 steps from the 3rd grade to the 9th grade. Cameron et al. [34] reported that the mean step count among 5−10-year-old Canadian children was 11,326 steps, but declined to 8488 steps among 15−19-year-old adolescents. Ischi et al. [35] reported corresponding declines in steps among Japanese children and adolescents. The decline in physical activity at the time when children turn to adolescents seems to be a global challenge [36]. 

The most active tertile took, on average, three times more steps than the least active tertile at all grades during the weekend. The mean step count of the least active tertile on the weekend was close to that reported among adult cardiovascular disease patients [37]. During school days, the difference was somewhat smaller, but the least active tertile still took about half the number of steps than the most active tertile. Further, among the 3rd and 5th graders, the mean number of steps varied during school hours, whereas the differences attenuated among the 7th and 9th graders. The difference between the tertiles was greatest during evening hours on weekdays. The most active tertile took on weekday evenings, on average, four times more steps per hour than the least active tertile. 

These FSPA study findings highlight that the children and adolescents were more active during weekdays than weekends. Particularly, the least active tertiles had the highest step count and the highest MET peak level during school hours. The results underline that the PA levels of the least active children and adolescents may benefit from daily school routines that include various forms of PA. Furthermore, there are notable differences in daily patterns between weekdays and weekends between the tertiles.

Quite little is known about the patterns of PA in children and adolescents, whereas the comparisons between studies remain challenging due to heterogeneous research methods, e.g., regarding the chosen outcomes and definitions of time segments. Steele et al. [14] investigated the volume and patterns of VPA across the segmented week (before and after school, school hours, and the weekend) to identify opportunities and target groups for the promotion of VPA. The children aged 9−10 years in the UK accumulated the most VPA before school, at break times, and after school. They did not find a difference in PA between weekdays and weekends, thus more SB was accumulated during the weekdays. Brooke et al. [38] described the changes in PA between ages 10 and 14 years using the same patterns as Steele et al. [14] with an extra lunch-time period, and found that the PA levels declined in all time segments when children got older, except in lesson-times. Total PA was greater on weekends than weekdays, and greater during out-of-school hours than during school hours. The findings from these previous PA pattern studies are somewhat in line with the present FSPA study. 

Few studies have employed tertiles or quartiles to explore daily MVPA patterns. Belton et al. [13] used quartiles and three specified periods for weekdays and weekends. Their findings suggested that the most active quartile of 12−14-year-old boys differed significantly from the two least active quartiles during every period except weekend mornings. In girls, the difference between the quartiles was smaller. The most significant difference between the most active quartile and all other quartiles was seen after school hours. Fairlough et al. studied [39] weekday–weekend differences between MVPA-quartiles among 9−10-year-old children in England. They reported that the three least active quartiles were significantly less active on weekends than on weekdays, and that the most active tertile maintained their PA levels consistently across the whole week. Garriquet and Colley [40] used tertiles to categorize MVPA among 6–19-year-old Canadians every two hours. In the younger children, the most active period was lunch-time (from 11 a.m. to 1 p.m.), and in adolescents, the after-school period (from 3 p.m. to 5 p.m.). Moreover, the children and adolescents were more active on weekdays than on weekends. In general, the PA pattern findings were quite similar to the present FSPA study results, indicating that, during school days, the most active hours are around lunch, and the least active pupils accumulate their PA mostly during school hours. On the other hand, the most active tertiles were more active during weekday leisure time, and children and adolescents were generally more active on weekdays than on weekends.

A major strength of the present FSPA study pertains to the yet largest population-based sample of Finnish children and adolescents, whose PA and SB were measured using accelerometers. The participants’ eagerness to wear the accelerometer among all four grades is also a strength. The participants wore the accelerometers, on average, for 13 h 49 min in the 3rd grade, and the wearing time increased up to 14 h and 14 min in the 9th grade. The wearing hours are in line with previous studies [5,7]. The high mean wearing time increases the representativeness of the data and the credibility of the findings. Further, validated MAD-APE algorithms based on raw triaxial accelerometer data used in the present study are device-independent, allowing better comparison in the future between different studies [22,23,24,25,26,27]. Moreover, the accelerometer-measured PA data are more robust than self-reported data [9,10,11]. 

There are also some limitations. First, this study describes only the results from accelerometer measurements of the FSPA study. The results of the simultaneously conducted FSPA survey were not utilized in the present analysis, but they have been and will be reported later elsewhere [41]. Without this survey data, linking the characteristics of the children and adolescents to physical activity patterns remained somewhat narrow. Second, the overall response rate of the accelerometer measurement was about 40%. 

Versatile and easily-implemented means would be needed to reduce sedentary time and promote PA in children’s and adolescents’ everyday life. The present findings indicate, parallel to the previous research, that the most active children and adolescents are more active, on average, because they accumulate more PA throughout the days [38,39,40,42]. Special attention should be paid to the children and adolescents who are not sufficiently active. Furthermore, our findings support tailored intervention strategies to certain periods, devised specifically for boys and girls [20,38,39,40,42]. There is a need for multilateral strategic cooperation and actions which extend across children’s and adolescents’ everyday life during school hours, but even more to leisure time, where the PA levels are low or decline rapidly [43]. Family is also important for the support and promotion of healthy movement behaviors of children and adolescents [44]. Earlier studies underline that children and especially adolescents should be involved in planning and creating these strategies and actions [45].

## 5. Conclusions

In the present FSPA study, there was great variation in children’s and adolescents’ PA levels and patterns between grades, sex, and the tertiles, especially during leisure time. Furthermore, the participants spent more than half of their waking hours sedentary or standing, SB increased significantly with grades, and boys were both more active and more sedentary compared to girls. The children and adolescents accumulated, on average, just under 10,000 steps per day, and they were more active on weekdays than weekends. On average, only one third of schoolchildren met the PA recommendation, and a higher proportion of lower-graders and boys met the recommendation more often than upper-graders and girls.

There is a need for targeted multi-level PA interventions among children and especially among adolescents to increase PA and reduce SB. Future research should focus on more detailed accelerometer-based measurements to identify the time segments of the week with the greatest potential to attain positive behavior changes among children and adolescents. 

## Figures and Tables

**Figure 1 ijerph-19-06950-f001:**
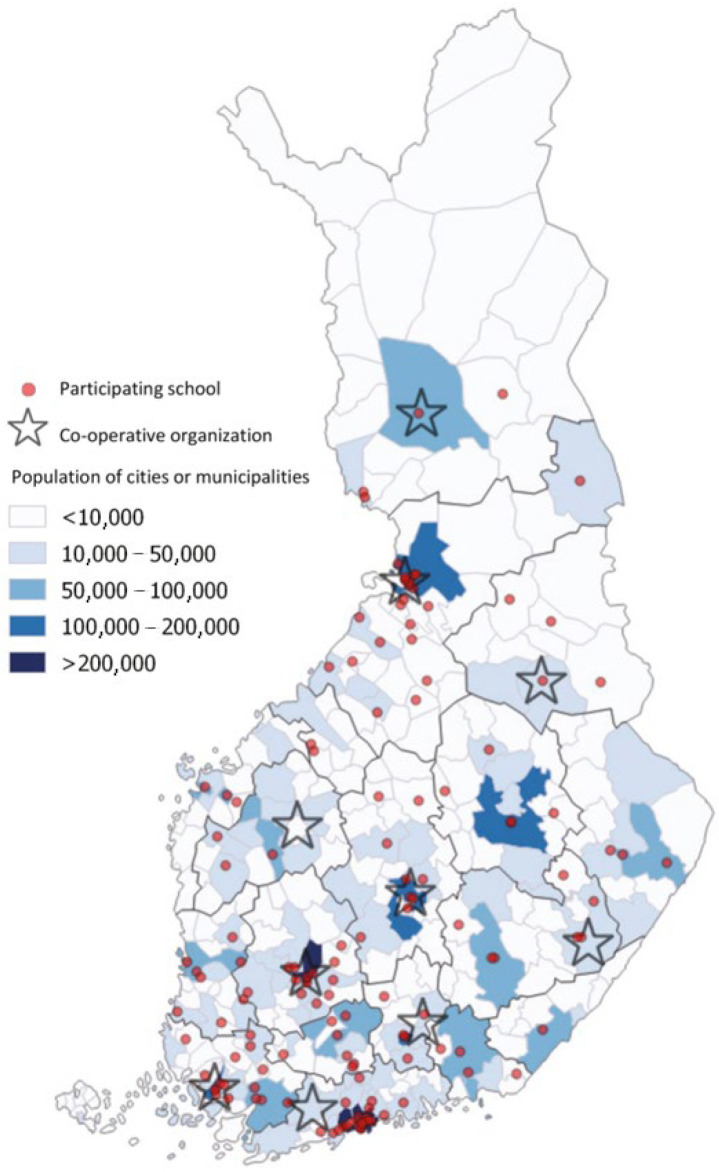
Regional distribution of co-operative organizations, cities, and municipalities.

**Figure 2 ijerph-19-06950-f002:**
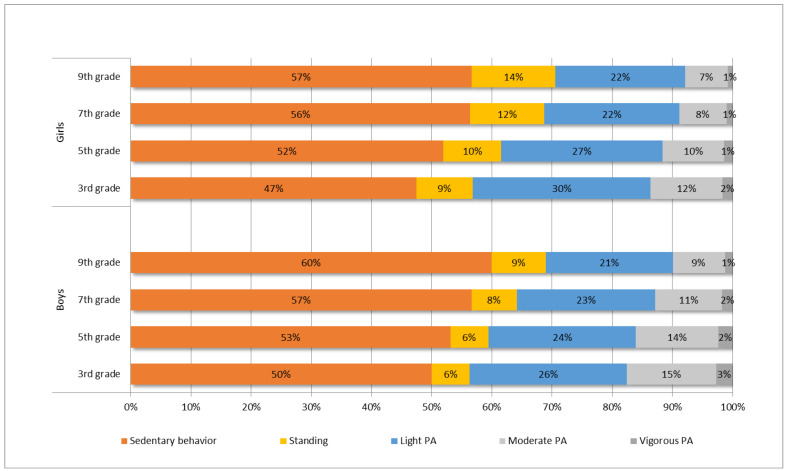
Proportion of participants’ physical activity and sedentary behavior by grade and sex.

**Figure 3 ijerph-19-06950-f003:**
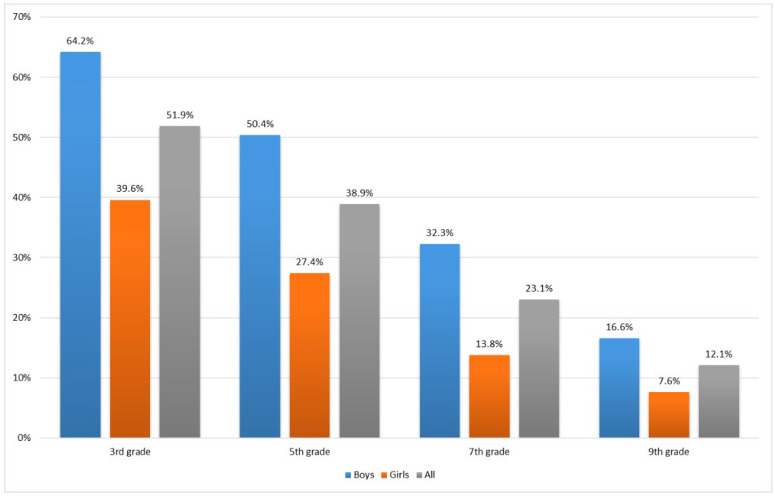
Participants’ meeting, on average, the physical activity recommendation.

**Figure 4 ijerph-19-06950-f004:**
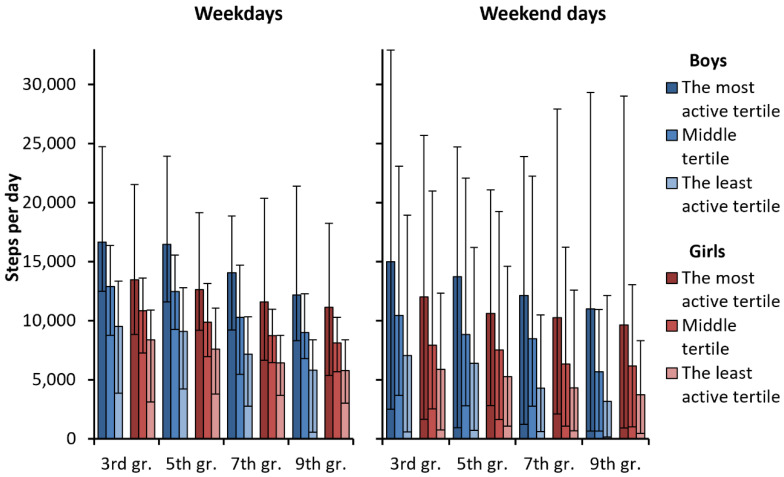
Participants’ number of steps, on average, by tertiles.

**Figure 5 ijerph-19-06950-f005:**
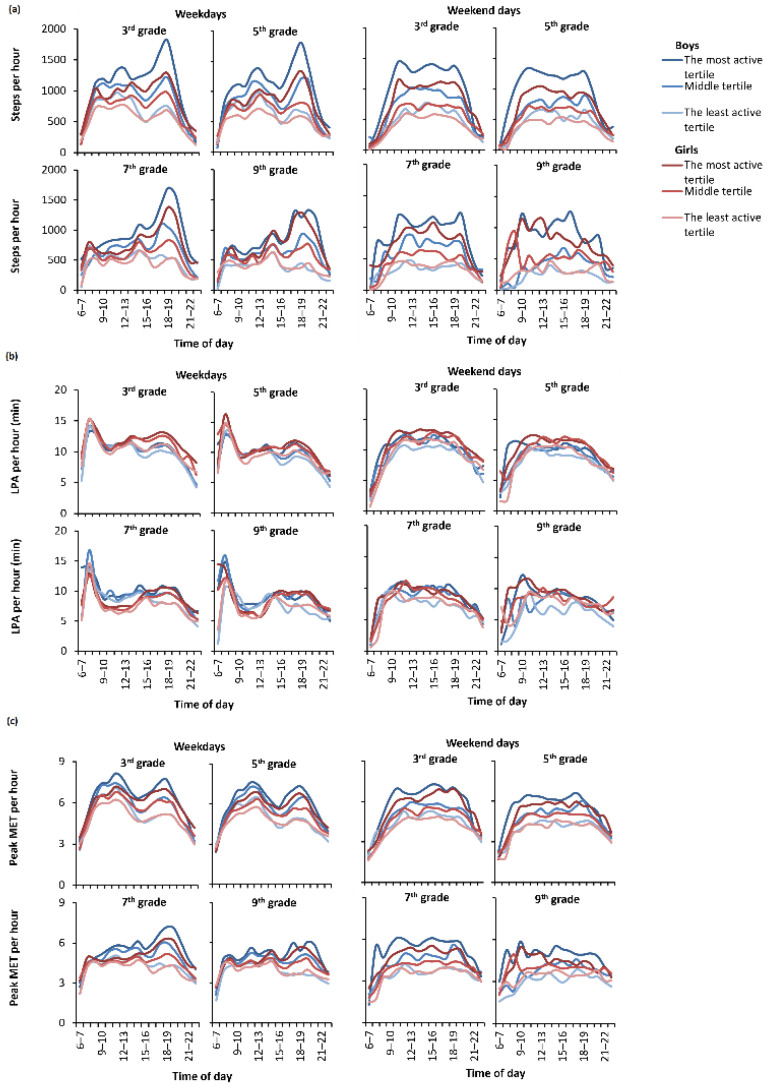
Participants’ hour-by-hour physical activity patterns by tertiles. Part (**a**) presents steps per hour, part (**b**) presents light physical activity per hour and part (**c**) presents peak MET per hour.

**Table 1 ijerph-19-06950-t001:** Participants’ accelerometer wearing days and wearing times.

**(A) The amount of the days (%) that 3274 participants used accelerometer by grade and sex**												
	**3rd grade**			**5th grade**			**7th grade**			**9th grade**		
**Wearing days**	**Boys**	**Girls**	**All**	**Boys**	**Girls**	**All**	**Boys**	**Girls**	**All**	**Boys**	**Girls**	**All**
1–3	11	7	8	6	5	5	15	10	12	18	8	12
4	6	8	7	9	9	9	11	6	8	10	11	10
5	13	15	14	17	18	18	17	17	17	17	17	17
6	28	27	28	25	29	27	27	25	26	22	26	24
7	43	43	43	43	39	41	29	42	37	33	39	37
n	441	536	977	443	519	962	302	480	782	214	339	553
**(B) The average wearing time of those 2981 participants who used the accelerometer at least four days and at least 10 h per day**												
	**3rd grade**			**5th grade**			**7th grade**			**9th grade**		
	**Boys**	**Girls**	**All**	**Boys**	**Girls**	**All**	**Boys**	**Girls**	**All**	**Boys**	**Girls**	**All**
h:min/day	13:52	13:46	13:49	14:02	13:55	13:58	14:07	14:16	14:13	14:18	14:12	14:14
n	394	500	894	416	495	911	257	431	688	176	312	488

**Table 2 ijerph-19-06950-t002:** Participants’ physical activity and sedentary behavior, standard deviation, and *p*-values for sex and grade differences.

	Grade	Boys Mean (sd)	Girls Mean (sd)	*p*-Value for SEX Difference ^¶^	All Mean (sd)	*p*-Value for Grade Difference ^¶,§^
Sedentary behavior (sitting, laying down) ^†^	3	6:56 (1:34)	6:32 (1:22)	<0.001	6:44 (1:28)	reference
	5	7:27 (1:41)	7:14 (1:28)	0.044	7:21 (1:34)	<0.001
	7	8:00 (1:32)	8:03 (1:25)	0.812	8:01 (1:28)	<0.001
	9	8:35 (1:40)	8:03 (1:24)	<0.001	8:19 (1:32)	<0.001
	All	7:42 (1:44)	7:30 (1:32)	0.002	7:27 (1:36)	
Standing ^†^	3	0:53 (0:28)	1:17 (0:32)	<0.001	1:05 (0:38)	reference
	5	0:53 (0:28)	1:20 (0:32)	<0.001	1:07 (0:30)	0.562
	7	1:04 (0:30)	1:46 (0:42)	<0.001	1:25 (0:36)	<0.001
	9	1:18 (0:38)	1:58 (0:51)	<0.001	1:38 (0:45)	<0.001
	All	1:01 (0:31)	1:36 (0:44)	<0.001	1:18 (0:41)	
Light PA ^†^	3	3:38 (0:35)	4:04 (0:38)	<0.001	3:51 (0:36)	reference
	5	3:26 (0:37)	3:44 (0:36)	<0.001	3:35 (0:37)	<0.001
	7	3:15 (0:43)	3:12 (0:37)	0.641	3:13 (0:40)	<0.001
	9	3:01 (0:50)	3:04 (0:42)	0.330	3:02 (0:46)	<0.001
	All	3:20 (0:44)	3:30 (0:45)	<0.001	3:31 (0:44)	
Moderate PA ^†^	3	2:03 (0:35)	1:38 (0:27)	<0.001	1:51 (0:31)	reference
	5	1:55 (0:33)	1:26 (0:24)	<0.001	1:40 (0:29)	<0.001
	7	1:34 (0:36)	1:07 (0:23)	<0.001	1:21 (0:29)	<0.001
	9	1:14 (0:33)	1:00 (0:23)	<0.001	1:07 (0:28)	<0.001
	All	1:43 (0:39)	1:17 (0:28)	<0.001	1:32 (0:36)	
Vigorous PA ^†^	3	0:23 (0:16)	0:14 (0:11)	<0.001	0:18 (0:14)	reference
	5	0:20 (0:16)	0:12 (0:10)	<0.001	0:16 (0:13)	<0.001
	7	0:15 (0:13)	0:08 (0:09)	<0.001	0:11 (0:11)	<0.001
	9	0:11 (0:11)	0:07 (0:07)	<0.001	0:09 (0:09)	<0.001
	All	0:17 (0:15)	0:10 (0:10)	<0.001	0:14 (0:13)	
Steps ^‡^	3	12,498 (3419)	10,345 (2532)	<0.001	11,421 (2975)	reference
	5	11,909 (3421)	9476 (2330)	<0.001	10,692 (2876)	<0.001
	7	10,034 (3225)	8483 (2529)	<0.001	9258 (2877)	<0.001
	9	8513 (3386)	7927 (2658)	0.027	8220 (3022)	<0.001
	All	10,824 (3743)	9040 (2669)	<0.001	9890 (3339)	

^†^ h:min, ^‡^ number, ^¶^ Mann–Whitney U-test, ^§^ Compared to 3rd graders (reference)

**Table 3 ijerph-19-06950-t003:** Tertile-wise daily step counts by grade and sex.

**Boys**				
	3rd grade	5th grade	7th grade	9th grade
the most active group	≥13,465	≥12,943	≥11,1889	≥9233
the middle group	13,464–10,950	12,942–10,056	11,188–8217	9232–6493
the least active group	≤10,949	≤10,055	≤8216	≤6492
**Girls**				
	3rd grade	5th grade	7th grade	9th grade
the most active group	≥11,058	≥10,100	≥9212	≥8419
the middle group	11,057–9048	10,099–8146	9211–7047	8418–6494
the least active group	≤9047	≤8145	≤7046	≤6493

## Data Availability

The FSPA study’s accelerometer data are maintained at the UKK Institute. The datasets analyzed in the present study are not publicly available due to ethical restrictions (the Ethics Committee of the University of Jyväskylä), but more detailed information on the data is available from the corresponding author on reasonable request.

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
