# Peer review of "Accelerometer-Measured Physical Activity Levels and Patterns Vary in an Age- and Sex-Dependent Fashion among Finnish Children and Adolescents"

_ijerph, 2022, doi:10.3390/ijerph19116950_

Round 1

Reviewer 1 Report

Once the review has been carried out, the research provides very relevant results of the physical activity carried out by schoolchildren, especially in the differentiation by school year and therefore between children and adolescents, and sex.

Perhaps the breadth of the results and objectives for the same paper is where it entails the difficulty in the possible deepening of them. Congratulations for the rigor of the work and the attempt to give a very complete vision of AF and sitting based on so many variables.

Suggestions:

In the title, include according to age and sex, since this differentiation is made in objectives and results.

Remove from the keywords those that already appear in the title.

In general, in the Introduction section, when the results of other studies are provided, for example regarding compliance with PA recommendations or daily step count, provide results in the two population segments studied: children and adolescents, not just in one of them.

In the methodology section, study design and sample subsection: provide more data on the type of study and specify specific data on the sample, specifying the number of subjects and percentages of children and adolescents (these data appear later in the results but should appear here ).

In the results section, remove references to methodological aspects that appear in lines 141-143 that refer to the sample and inclusion criteria for the use of accelerometers to include them in the methodology section.

In some of the written results, references to differentiated percentages between children and adolescents are included. For example, on line 178, the general result appears: 31.3% of the participants comply with the recommendations. Why not try to provide the data on the percentage of children and adolescents, for example, by grouping subjects from third and fifth grade and on the other hand those from 7th to 9th grade?

Again, my congratulations

Author Response

Comments and Suggestions for Authors

Once the review has been carried out, the research provides very relevant results of the physical activity carried out by schoolchildren, especially in the differentiation by school year and therefore between children and adolescents, and sex.

Perhaps the breadth of the results and objectives for the same paper is where it entails the difficulty in the possible deepening of them. Congratulations for the rigor of the work and the attempt to give a very complete vision of AF and sitting based on so many variables.

Response:
Thank you for these encouraging comments
.

Suggestions:

In the title, include according to age and sex, since this differentiation is made in objectives and results.

Response:
The title has been modified as suggested:

“Accelerometer-measured physical activity levels and patterns vary in an age- and sex-dependent fashion among Finnish children and adolescents”

Remove from the keywords those that already appear in the title.

Response:
Thank you for pointing this out. We have removed the keywords that already appear in the title (“accelerometer”, “children”, “adolescents”, “physical activity patterns”) and added a few new ones: “daily steps”, “moderate intensity”, “light intensity”.

In general, in the Introduction section, when the results of other studies are provided, for example regarding compliance with PA recommendations or daily step count, provide results in the two population segments studied: children and adolescents, not just in one of them.

Response:
This more detailed information about PA recommendation with children and adolescents has been revised as suggested in the beginning of the Introduction:

According to accelerometer-based measurements, most of the children’s and adolescents’ PA consists of light physical activity (LPA) [4, 5]. Approximately half of children and less than one-quarters of adolescents meet the PA recommendation [1, 6]; i.e., at least 60 minutes moderate to vigorous physical activity (MVPA) in a day [4, 5, 6, 7, 8].

In step counts we would suggest that  we keep on average perspective due to large variation of the research results in different studies.

In the methodology section, study design and sample subsection: provide more data on the type of study and specify specific data on the sample, specifying the number of subjects and percentages of children and adolescents (these data appear later in the results but should appear here ).

Response:
This information from lines 139-148, including Table 1. has been moved to the methodology section as suggested.

In the results section, remove references to methodological aspects that appear in lines 141-143 that refer to the sample and inclusion criteria for the use of accelerometers to include them in the methodology section.

Response:
This has been revised as suggested. We came to a conclusion that the whole Participants section should be stated under the Methodology headline. Including the Table 1.

In some of the written results, references to differentiated percentages between children and adolescents are included. For example, on line 178, the general result appears: 31.3% of the participants comply with the recommendations. Why not try to provide the data on the percentage of children and adolescents, for example, by grouping subjects from third and fifth grade and on the other hand those from 7th to 9th grade?

Response:
Thank you for this valuable comment. The results have been revised accordingly:

According to accelerometer measurements, on average one-third (31.3%) of the FSPA study participants met the PA recommendation accumulating at least 60 minutes of MVPA each day. 45,4% of children (3rd and 5th graders) and 18,6% of adolescents (7th and 9th graders) met the PA recommendation. Girls were less likely to meet the recommendation than boys. (Figure 3.)

Again, my congratulations

Response:
Thank you very much!

Reviewer 2 Report

 Dear Athors 
Article  "Accelerometer-measured physical activity patterns vary in an age-dependent fashion among Finnish children and adolescents" is interesting and have valuable  scientific content, new data and  corect statistics. 
I would like to kindly ask you to add in References the article entitled: Wasik J, Ortenburger D, Gora T, Mosler D. The influence of effective distance on the impact of a punch - Preliminary Analysis. Physical Activity Review 2018; 6: 81-86. doi: 10.16926/par.2018.06.11  Published online: 9.05.2018 
The proposed quotation of the following work  in References  are the result of meritorical reasons. The proposed work for quotation is closely related to the subject matter of the paper under review, which includes significant and relevant new information. With relation to the fact that the authors of the paper under review may not have been familiar with this paper, I indicated this in the review as a quotation in order to be of benefit to the your very valuable  paper.

Yours faithfully 
Reviewer

Author Response

Comments and Suggestions for Authors

Dear Athors 
Article  "Accelerometer-measured physical activity patterns vary in an age-dependent fashion among Finnish children and adolescents" is interesting and have valuable scientific content, new data and corect statistics. 

Response:
Thank you very much. We highly appreciate your comments.

I would like to kindly ask you to add in References the article entitled: Wasik J, Ortenburger D, Gora T, Mosler D. The influence of effective distance on the impact of a punch - Preliminary Analysis. Physical Activity Review 2018; 6: 81-86. doi: 10.16926/par.2018.06.11  Published online: 9.05.2018 
The proposed quotation of the following work  in References  are the result of meritorical reasons. The proposed work for quotation is closely related to the subject matter of the paper under review, which includes significant and relevant new information. With relation to the fact that the authors of the paper under review may not have been familiar with this paper, I indicated this in the review as a quotation in order to be of benefit to the your very valuable  paper.

Response:
Thank you for pointing this interesting paper out. We familiarized ourselves with the Wasik et al. article, but we did not find a proper place for it in our revised manuscript due to the older age group (young adults) and specific sports activity (martial arts).

Reviewer 3 Report

Dear authors,

Thank you for this nice manuscript presenting data on physical activity (PA), sedentary 10 behavior (SB), and hour-by-hour PA patterns among Finnish children and adolescents. 

I have some minor comments and a major concern which are presented below.

Minor comments:

  • The title of the manuscript does not match the main objective of the study. Moreover, it might not be the main result you present (at least it was not highlighted along with the manuscript). I do recommend changing it in order to avoid the wrong expectations.
  • I also recommend adjusting conclusions (both in the abstract and in the main text). It should answer the objectives. Part of the conclusions in the main text can be addressed along with the discussion (as it already is).

Major concern:

My major concern is regarding statistical analysis. You have a huge sample, and comparisons between boys and girls and among grades were performed through the Mann-Whitney test. By doing that you have increased a lot of your statistical power. However, once you have multiple comparisons, adjustments in the significance level are required and you have done that by using the Benjamini-Hochberg -procedure. I am not a specialist in statistics but it seems much more feasible to use a single test (such as a  two-way ANOVA) to compare both factors (sex and grades) and present the multiple comparisons. I did not feel comfortable with the results once I am not secure about the analysis.

Author Response

Comments and Suggestions for Authors

Dear authors,

Thank you for this nice manuscript presenting data on physical activity (PA), sedentary behavior (SB), and hour-by-hour PA patterns among Finnish children and adolescents. 

I have some minor comments and a major concern which are presented below.

Minor comments:

  • The title of the manuscript does not match the main objective of the study. Moreover, it might not be the main result you present (at least it was not highlighted along with the manuscript). I do recommend changing it in order to avoid the wrong expectations.

Response:
The title has been modified to better comply with the main objective of the study: "Accelerometer-measured physical activity levels and patterns vary in an age- and sex-dependent fashion among Finnish children and adolescents"

  • I also recommend adjusting conclusions (both in the abstract and in the main text). It should answer the objectives. Part of the conclusions in the main text can be addressed along with the discussion (as it already is).

Response:
Thank you for this comment. The conclusions have been adjusted both in the Abstract as well as in the Main text.

Major concern:

My major concern is regarding statistical analysis. You have a huge sample, and comparisons between boys and girls and among grades were performed through the Mann-Whitney test. By doing that you have increased a lot of your statistical power. However, once you have multiple comparisons, adjustments in the significance level are required and you have done that by using the Benjamini-Hochberg -procedure. I am not a specialist in statistics but it seems much more feasible to use a single test (such as a  two-way ANOVA) to compare both factors (sex and grades) and present the multiple comparisons. I did not feel comfortable with the results once I am not secure about the analysis.

Response:
Thank you for this comment.

All statistical analyses were carried out by statistician who is specialised in population-based studies in PA and SB.

The main reason for us to use non-parametric Mann-Whitney test was really skewed distribution of vigorous PA. Our sample size is indeed so large and the differences between most of the mean SB and PA times and steps counts are so huge that we have plenty of statistical power to find significant differences. We don’t feel that the selected statistical method is the reason for significant findings. Results are logical and it is easy to believe that the significant differences we found are real (mean differences are big and selected statistical model doesn’t change that). We also feel that Mann-Whitney for sub-analysis (for example we always compare one of 5th, 7th or 9th to 3rd, not all against all) with Benjamini-Hochberg correction significantly adds power compared to using single two-way ANOVA or ANCOVA for each outcome to compare multiple comparison adjusted marginal means. We agree that ANCOVA could add power to our analysis for normally distributed outcomes.

However, we did some additional analysis to see how the results would change with ANCOVA. For all the highly significant p-values at table 2 (p<0.001) p-values would remain the same when using Bonferroni-corrected p-values from ANCOVA. For the borderline p-values (sex difference of SB on 5th graders and sex difference of steps on 9th graders) Bonferroni corrected marginal means from ANCOVA are p=0.030 (Mann-Whitney p=0.044) and p=0.012 (p=0.027). Therefore, changing the statistical approach does not really change the statistical significance of the presented results.
